# Chatting Makes Perfect: Chat-based Image Retrieval

**Matan Levy**[1]        **Rami Ben-Ari**[2]        **Nir Darshan**[2]        **Dani Lischinski**[1]

[1]The Hebrew University of Jerusalem, Israel
[2]OriginAI, Israel
Levy@cs.huji.ac.il

## Abstract

Chats emerge as an effective user-friendly approach for information retrieval, and are successfully employed in many domains, such as customer service, healthcare, and finance. However, existing image retrieval approaches typically address the case of a single query-to-image round, and the use of chats for image retrieval has been mostly overlooked. In this work, we introduce ChatIR: a chat-based image retrieval system that engages in a conversation with the user to elicit information, in addition to an initial query, in order to clarify the user's search intent. Motivated by the capabilities of today's foundation models, we leverage Large Language Models to generate follow-up questions to an initial image description. These questions form a dialog with the user in order to retrieve the desired image from a large corpus. In this study, we explore the capabilities of such a system tested on a large dataset and reveal that engaging in a dialog yields significant gains in image retrieval. We start by building an evaluation pipeline from an existing manually generated dataset and explore different modules and training strategies for ChatIR. Our comparison includes strong baselines derived from related applications trained with Reinforcement Learning. Our system is capable of retrieving the target image from a pool of 50K images with over 78% success rate after 5 dialogue rounds, compared to 75% when questions are asked by humans, and 64% for a single shot text-to-image retrieval. Extensive evaluations reveal the strong capabilities and examine the limitations of CharIR under different settings. Project repository is available at https://github.com/levymsn/ChatIR.

## 1   Introduction

Users have always been the central focus of information retrieval. Conversational search offers opportunities to enhance search effectiveness and efficiency. The tremendous growth in the volume of searchable visual media underscores the need for fast and reliable retrieval systems. Retrieval capabilities are indispensable in general internet image search, as well as in specific domains, such as e-commerce or surveillance. Current approaches to image retrieval in computer vision primarily focus on image-to-image [10, 46], text-to-image [30, 31] and composed-image retrieval [19, 27]. However, a single query might fail to fully convey the search intent, and multiple trials may be required before a satisfactory result is retrieved. Furthermore, it is up to the user to decide how to modify the query in each trial, while the retrieval system processes each attempt independently.

Motivated by these difficulties and inspired by recent progress in Large Language Models (LLM), which have demonstrated unprecedented natural language chat capabilities [36–38, 48], we introduce and explore a new image retrieval "protocol": Chat-based Image Retrieval, which we dub ChatIR. A schematic view of ChatIR and the system that we propose in this paper is provided in Figure 1. The process starts with a user-provided short *description* of the desired image, similarly to text-to-image retrieval. However, from this point on, the retrieval system is able to progressively refine the query by

37th Conference on Neural Information Processing Systems (NeurIPS 2023).

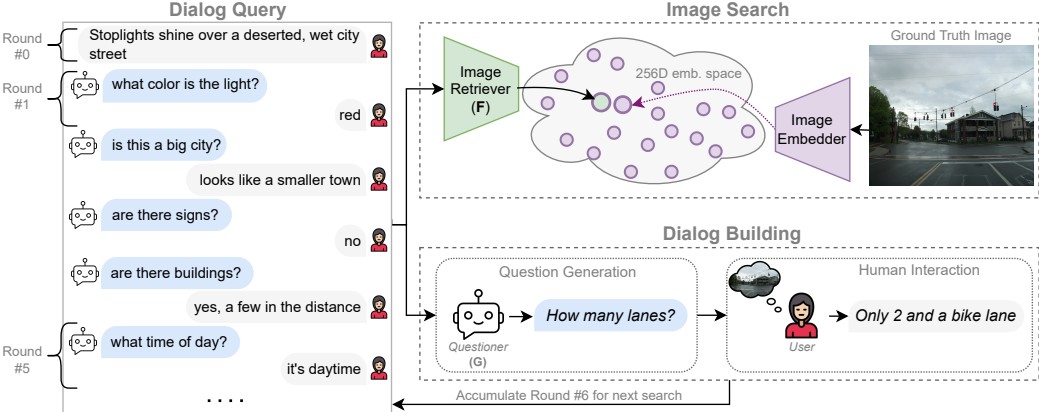

Figure 1: An overview of Chat Image Retrieval. The pipeline consist of two stages: Image Search (IS) and Dialog Building (DB). The IS stage takes as an input the ongoing dialog, composed of image caption and a few rounds of Q&As, in order to find the target image. Note that a dialog of length 0 is solely the image caption, equivalent to Text-to-Image retrieval task. The DB stage provides the follow-up question to the current dialog.

actively polling the user for additional information regarding the desired result. Ideally, a ChatIR system should avoid gathering redundant information, and generate dialogues that steer it towards the desired result as quickly as possible. Note that this gradual progress scenario is very different and more natural from providing at the outset an overly descriptive caption, which hypothetically contains all of the required information. In contrast, ChatIR proactively obtains the information from the user and is able to process it in a unified and continuous manner in order to retrieve the target image within a few question answering (Q&A) rounds.

Specifically, the ChatIR system that we propose in this work consists of two stages, Image Search (IS) and Dialog Building (DB), as depicted in Figure 1. Image search is performed by an image retriever model $F$, which is a text encoder that was trained to project dialogues sequences (of various lengths) into a visual embeddings space. The DB stage employs a question generator $G$, whose task is to generate the next question for the user, taking into account the entire dialog up to that point. The two components of ChatIR are built upon the strong capabilities of *instructional* LLMs (where the model is instructed about the nature of the task) and foundation Vision and Language (V&L) models.

Addressing this task we are faced with three main questions: 1. What dataset do we use and is it necessary to create and manually label such a dataset? 2. How do we independently evaluate different components of the system? 3. How do we define a benchmark and a performance measure for this task to make further progress in this domain measurable?

To mitigate the cumbersome and costly process of collecting human-machine conversations we use the *VisDial* dataset [8]. Although this dataset was designed and generated to create chats about images without any retrieval goal, we use the specific image related to each dialog as our retrieval target, and the dialog as our chat. In our case the questioner is an agent and the answerer is a human while in the Visual Dialog task [8] it is vice versa.

Considering the goals of ChatIR as a conversational retrieval system, we evaluate its performance by measuring the probability of a successful retrieval up to each round in the dialog. We use this metric to systematically study the major components of the framework and to examine the impact of different questioner models $G$, and training strategies for $F$ on retrieval performance.

For example, when training $F$ using various masking strategies, we found that masking the initial descriptions proved to be the most effective method (elaborated in Section 5). Since the retrieval performance of ChatIR also depends on the quality of the questions generated by $G$, we evaluate several alternatives for $G$ based on their impact on $F$'s retrieval ranking. One of the problems in this evaluation is the need for a user in the loop, to answer $G$'s questions (at inference time), while taking into account the chat history. Such evaluation of ChatIR is obviously costly and impractical at scale. To mitigate this, we replace the user with a multi-purpose vision-language model BLIP2 [21], as a

*Visual Dialog Model* (VDM) that answers questions. We further harvest human answers testing our system in the real scenario with a human providing the answers, and show a comparison between the VDM and humans in terms of impact on the performance of ChatIR.

We find that ChatIR can retrieve the target image from a corpus of 50K images, within the top-10 results, with success rate of 78.3% and 81.3%, after 5 and 10 Q&A rounds, respectively. Overall, ChatIR increases retrieval success by 18% over a single-shot text-to-image retrieval where the user provides only a text description.

In summary, our contributions are as follows:

- Introduction of ChatIR, a novel framework for visual content search guided by an interactive conversation with the user.
- We explore the ChatIR idea leveraging foundation V&L models, with several LLM questioners and image retrieval training strategies.
- We suggest evaluation protocols suitable for continual progress and assessment of questioners using a Visual Dialog model in place of a human.
- We test our framework on real human interactions by collecting answers from users and further evaluate our method against strong baselines generated from prior art.

## 2   Related Work

Man-machine dialogue has been used for information retrieval for decades [35]. More recent works involving chatbots for large knowledge corpus include [11, 28, 39]. Below we survey related tasks that involve visual modality and dialogues.

**Visual Conversations:**   In visual domain, there are many applications that cope with only one-step human-image interactions, *e.g.*, VQA [1, 13, 29, 50, 56], Image Retrieval (in its variations) [2, 22, 23, 44, 56], and Composed Image Retrieval [2, 4, 12, 16, 49]. While the output of these methods are either images or answers, Visual Question Generation tackles a counter VQA task, and tries to generate a single question about the image [25, 42]. Visual Dialog [8, 33] is the task of engaging in a dialog about an image, where the user is the questioner and the machine is the answerer that has access to the image. With the bloom of text-based image generation models, some approaches suggest to initiate a chat for image generation. For instance, Mittal *et al*. [32] propose a method to generate an image incrementally, based on a sequence of graphs of scene descriptions (scene-graphs). Wu *et al*. introduced *VisualChatGPT* [51], an "all-in-one" system combining the abilities of multiple V&L models by prompting them automatically, for processing or generating images. Their work focuses on access management for different models of image understanding and generation. A concurrent work [59] uses ChatGPT in conjunction with BLIP2 [21] to enrich image captioning. Although all the above studies deal with combinations of vision and language, none of them target image retrieval.

A slightly different line of work addresses the problem of generating dialogues about images, called *Generative Visual Dialogues* [9, 34, 54, 57]. They generally focus on training two agents, Questioner-bot (Q-bot) and Answerer-bot (A-bot), in order to generate the dialog. The idea is to test the ability of machines in generating a natural conversation about an image. To this end, both bots had access to the dialog history while the A-bot can further "see" the image, for providing the answers. The training strategy is based on Reinforcement Learning (RL) while for evaluation an auxiliary task is used, named "Cooperative Image Guessing Game" with a reward where the Q-bot should predict the image in a pool of $\sim$ 9.5K images. However, the recent foundation V&L models outperform these methods in all aspects *e.g*. question answering [33] and question diversity (see also Section 4) and are shown to be effective also for various downstream tasks [5, 15, 19, 20, 22–24, 29, 33, 44, 50, 53, 56, 58]. Motivated by these capabilities, we build our model on LLM and V&L foundation models to create our ChatIR system. We compare our method to the related prior art in [9, 34], although these tasks are different from ours and do not directly target image retrieval.

**Visual Search:**   An important aspect of information retrieval is visual search and exploration. Involving the user in search for visuals is a longstanding task of Image Retrieval that has been previously studied by combining human feedback [14, 17, 18, 52]. The feedback types vary from a binary choice (relevant/irrelevant) *e.g.* [43, 45] through a pre-defined set of attributes [18, 40],

and recently by open natural language form, introduced as Composed Image Retrieval (CoIR) [19, 27, 49, 52]. The CoIR task involves finding a target image using a multi-modal query, composed of an image and a text that describes a relative change from the source image. Some studies construct a dialog with the user [14, 52] by leveraging such CoIR models where the model incorporates the user's textual feedback over an image to iteratively refine the retrieval results. However, these particular applications differ from ChatIR in not involving user interaction through questions, nor does it explicitly utilize the history of the dialogue. One way or another, this type of feedback requires the user to actively describe the desired change in an image, time after time without relation to previous results, as opposed to ChatIR where the user is being pro-actively and continually questioned, with including all the history in each search attempt.

## 3 Method

We explore a ChatIR system comprising two main parts: *Dialog Building (DB)* and *Image Search (IS)*, as depicted in Figure 1. Let us denote the ongoing dialog as $D_i := (C, Q_1, A_1, ..., Q_i, A_i)$, where $C$ is the initial text description (caption) of the target image, with $\{Q_k\}_{k=1}^i$ denoting the questions and $\{A_k\}_{k=1}^i$ their corresponding answers at round $i$. Note that for $i = 0$, $D_0 := (C)$, thus the input to IS is just the caption, *i.e.*, a special case of the Text-to-Image Retrieval task.

**Dialog Builder Model** The dialog building stage consists of two components, the Question generator $G$ and the Answer provider, which in practice is a human (the user) who presumably has a mental image of the target $T$. In our case $G$ is an LLM that generates the next question $Q_{i+1}$ based on the dialog history $D_i$, *i.e.* $G : D_i \rightarrow Q_{i+1}$. We assume that $G$ operates without the benefit of knowing what the target $T$ is. In this paper, we examine various approaches for the questioner $G$, exploring the capabilities and limitations as well as their failure modes (reported in Section 4). In order to enable experimenting with these different alternatives at scale, we cannot rely on user-provided answers to the questions proposed by $G$. Thus, in these experiments, all of the questions are answered using the same off-the-shelf model (BLIP2 [21]). A smaller scale experiment (reported in Section 4.3) evaluates the impact of this approach on the performance, compared to using human-provided answers.

**Image Retriever Model:** Following common practice in image retrieval [15, 19, 21–24, 27, 44, 49, 52], our IS process searches for matches in an embedding space shared by queries and targets (see Figure 1). All corpus images (potential targets) are initially encoded by an Image Embedder module, resulting in a single feature representation per image $f \in \mathbb{R}^d$, with $d$ denoting the *image* embedding space dimension. Given a dialog query $D_i$, the Image Retriever module $F$, a transformer in our case, maps the dialog $F : D_i \rightarrow \mathbb{R}^d$ to the shared embedding space. The retrieval candidates are ranked by cosine-similarity distance w.r.t the query embedding. As our $F$ we use BLIP [22] pre-trained image/text encoders, fine-tuned for dialog-based retrieval with contrastive learning. We leverage the text encoder self-attention layers to allow efficient aggregation of different parts of the dialog (caption, questions, and answers), and for high level perception of the chat history. Motivated by previous work [20, 29, 33], we concatenate $D_i$'s elements with a special separating token [*SEP*], and an added [*CLS*] token to represent the whole sequence. The latter is finally projected into the image embedding space.

We train $F$ using the manually labelled VisDial [8] dataset, by extracting pairs of images and their corresponding dialogues. We train $F$ to predict the target image embedding, given a partial dialog with $i$ rounds $D_i$, concatenating its components (separated with a special [SEP] token) and feeding $F$ with this unified sequence representing $D_i$. In Section 5 we demonstrate that randomly masking the captions in training boosts the performance.

**Implementation details:** we set an *AdamW* optimizer, initializing learning rate by $5 \times 10^{-5}$ with a exponential decay rate of 0.93 to $1 \times 10^{-6}$. We train the Image Retriever $F$ on VisDial training set with a batch size of 512 for 36 epochs. The Image Embedder is frozen, and is not trained. Following previous retrieval methods [19, 41], we use the Recall@K surrogate loss as the differentiable version of the Recall@K metric. Training time is 114 seconds per epoch on four *NVIDIA-A100* nodes. In testing, we retrieve target images from an image corpus of $50,000$ unseen COCO [26] images.

# 4 Evaluation

In this section we examine various aspects of ChatIR. We start by showing the benefit of ChatIR over existing Text-To-Image (TTI) retrieval methods. We conduct experiments on established TTI benchmarks (Flickr30K [55] and COCO [26]), and then proceed to evaluate our model on the human-annotated dialogue dataset (VisDial). Next, we compare ChatIR performance on VisDial using various Questioner models ($G$). Finally, we evaluate our top performing model with real humans as answerers, on a small validation subset. In all experiments, a large search space is used (50K images for VisDial, 30K for Flickr30K, and 5K for COCO).

## 4.1 Comparison With Existing Text-to-Image Retrieval Methods

Here we compare the retrieval performance of ChatIR with existing Text-To-Image methods. First, we generate two synthetic image-dialogue datasets (using ChatGPT as a questioner, and BLIP2 [21] as an answerer) from the two established TTI benchmarks: Flickr30K and COCO. In Figure 2 we compare our method to two TTI methods, CLIP and the publicly available SoTA baseline for TTI, BLIP [22], in a zero-shot setting (*i.e.*, none of the compared methods have been fine-tuned on either dataset). We find that our method surpasses the two baselines by a large margin, on both datasets. Furthermore, when we provide the baselines with the concatenated text of the dialogues, instead of just a caption, they exhibit a significant improvement over the single-hop TTI attempt. Nevertheless, the gap in favor of our method is maintained (Fig. 2a) or increased (Fig. 2b). These zero-shot results show that: 1) dialogues improve retrieval results for off-the-shelf TTI models. Although the CLIP and BLIP baselines have only been trained for retrieval with relatively short (single-hop) text queries, they are still capable of leveraging the added information in the concatenated Q&A text. Note that CLIP becomes saturated at a certain point due to the 77 token limit on the input. 2) Our strategy of training an Image Retriever model with dialogues (as described in Sec. 3) further improves the retrieval over the compared methods, raising accuracy from 83% to 87% at single-hop retrieval, and surpassing 95% after 10 dialogue rounds (on COCO). Next, we fine-tune SoTA TTI BLIP on VisDial (by providing it with images and their captions only) and compare it to our method on the VisDial validation set. Results presented in Figure 3. We make two main observations: 1) The retrieval performance of the fine-tuned single-hop TTI baseline of BLIP, is nearly identical to our dialogue-trained model (63.66% vs. 63.61%). This corresponds to dialogues with 0 rounds in Figure 3a. 2) Using a dialogue boosts performance, while increasingly longer dialogues with ChatIR eventually achieve retrieval performance over 81% (Fig. 3a), showing a significant improvement a single-hop TTI.

## 4.2 Comparison Between Questioners

We examine various Questioner models ($G$) and their relations with $F$ by evaluating the entire system. As previously discussed, we use a Visual Dialog (VD) model to imitate the user by answering $G$'s questions. More specifically, we use BLIP2 that was previously showed to be effective in zero-shot VQA and VD [7, 21]. Using the same VD model as answerer in our experiments allows a fair comparison between different questioner models. As retrieval measure we compute the rate of images that were successfully retrieved among the top-$k$ ranked results up to each dialog round (we opt for $k = 10$ here). We stop the chat for each image as soon as it reaches the top-$k$ list and add it to our success pool, since in practice, the user will stop the search at this stage. We examined the following LLM models for $G$ in our experiments:

**Few-Shot Questioner:** We test four pre-trained LLMs to generate the next question, based on few-shot instructions [3]. We explicitly instruct the model with the prompt *"Ask a new question in the following dialog, assume that the questions are designed to help us retrieve this image from a large collection of images"* alongside with a few train examples (shots) of dialogues with the next predicted question. Examples for such prompts can be found in our supplementary material. Specifically, we tested FLAN-T5-XXL [6], FLAN-ALPACA-XL [47], FLAN-ALPACA-XXL, and ChatGPT [36]. The examples in Figures 5 and 6 were generated using ChatGPT as questioner.

**Unanswered Questioner:** in this method we use an LLM to generate 10 questions at once, based solely on the given caption and without seeing any answers. Thus, although the actual retrieval is conducted using the full dialogues (questions and answers), the questions generated in this manner are not affected by the answers. Here we provide ChatGPT with the following prompt: *"Write 10 short questions about the image described by the following caption. Assume that the questions*

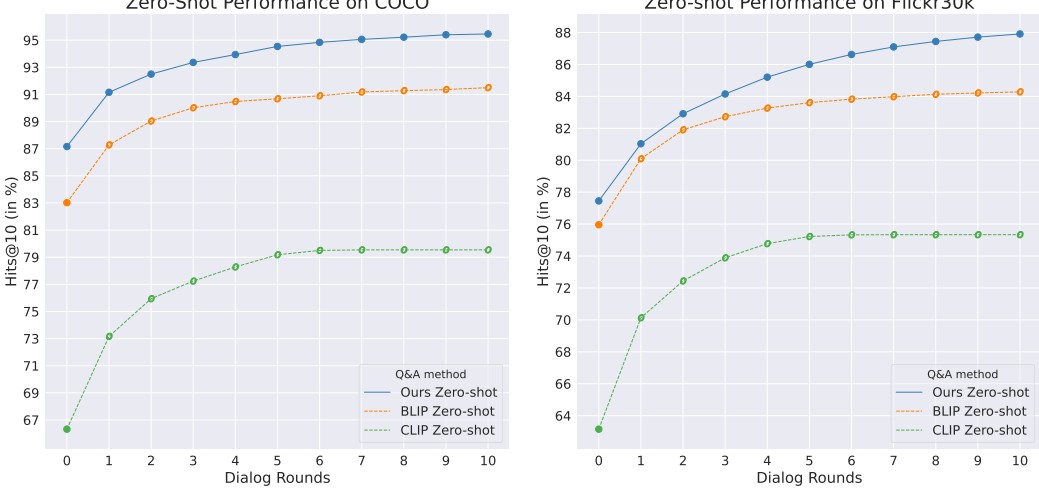

(a) Hits@10 success on COCO (higher is better).  (b) Hits@10 success on Flickr30k (higher is better).

Figure 2: In ChatIR, retrieval is attempted after every Q&A round, while in traditional TTI retrieval, there is a single attempt, without any Q&A involved (the leftmost dot in the green and orange curves). To examine whether the TTI baselines would benefit from the extra information conveyed in the dialogue, we also plot (as hollow points) retrieval attempts made by these baselines using a concatenation of increasing numbers of Q&A rounds. It may be seen that these concatenated queries improve the retrieval accuracy, even though CLIP and BLIP were not trained with such queries.

*are designed to help us retrieve this image from a large collection of images: [CAPTION]*". This experiment demonstrates how the influence of the answers on the question generation affects the retrieval performance.

**Human:** Here we use the human-labelled VisDial dataset [8]. We extract the questions from each dialog to simulate a human question generator.

In Figure 3 we present the performance of different Questioners with the same Image Retriever $F$. While Figure 3a presents the retrieval success rate (Hit rate), Figure 3b presents per-round performance in terms of Average Target Rank (ATR), where lower is better. The first observation shows a consistent improvement of retrievals with the length of the dialog, demonstrating the positive impact of the dialog on the retrieval task. Looking at the top-performing model, we already reach a high performance of 73.5% Hit rate in a corpus of 50K unseen images, after just 2 rounds of dialog, a 10% improvement over TTI (from $\sim 63\%$, round 0 in the plot). We also observe that questioners from the previous work of [9, 34] based on RL training are among the low performing methods.

Next, we see a wide performance range for FLAN models with FLAN-ALPACA surpassing human questioners in early rounds, while their success rate diminishes as the dialog rounds progress, causing them to underperform compared to humans. However, the success rate the ChatGPT (Unanswered) questioner (pink line), that excludes answer history, is comparable to that of humans, with less than $\sim 0.5\%$ gap. By allowing a full access to the chat history, the ChatGPT questioner (blue line) surpasses all other methods, mostly by a large margin (with $\sim 2\%$ over Human). Perhaps more importantly, in both ChatGPT questioners we see a strong and almost monotonic decrease in Average Target Rank (Fig. 3b) as the dialog progresses (blue and pink lines), similarly to the Human case (green dashed line). Other questioners fail to provide progressive improvement of the target image rank (lower ATR) with the dialog length, implying saturation of the relevancy of their questions. Interestingly, as dialogues progress, only the human questioners consistently lower the target rank to a minimum of almost $25/50,000$, demonstrating highest quality of questions.

Next, we measure question repetitions for each questioner as a suspected cause behind the performance levels of different questioners (previously addressed in [34]). For each questioner we calculate the average number of exact repetitions of questions per dialog (*i.e.*, out of 10). While the average number of repetitions is nearly 0 for either human or ChatGPT questioners, the other methods exhibit an average of $1.85 - 3.44$ repeated questions per dialog. We observe that these findings are correlated

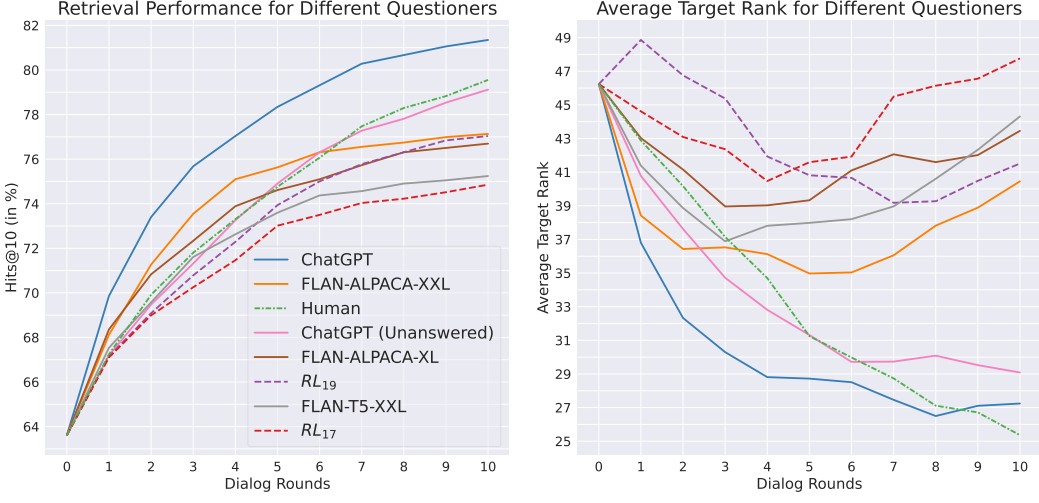

| (a) Questioners Hits@10 success (higher is better). | (b) Target image average rank (lower is better). |

Figure 3: **Left**: Evaluation of different chat questioner methods on VisDial. For all cases (including "Human") the answers are obtained from BLIP2. Note that dialog with 0 rounds is only the image caption, a special case of the text-to-image retrieval task. Top and super-human performance is obtained by ChatGPT with a significant gap over several versions of FLAN and previous RL based methods, $RL_{17}$ [9] and $RL_{19}$ [34]. **Right**: Average rank of target images after each round of dialog.

with the overall performance of questioners in Figure 3, since repetitive questions add little or no information about the target image. More details are available in our suppl. material, where we also measure uniqueness at the token level (repeated questions result in fewer unique tokens per-dialog).

## 4.3 Comparison to Human In The Loop

While above we examined all the questioner methods in conjunction with BLIP2 answers, here we evaluate the performance with as human as the answer provider. Our incentive for this experiment is 1) To evaluate how answers of BLIP2 compare to those of humans. Is there any domain gap? 2) To test our top-performing questioner in a real ChatIR scenario. To this end, we conduct dialogues on $\sim 8\%$ of the images in the VisDial [8] validation set (through a designed web interface), between ChatGPT (Questioner) and Human (Answerer). Note we also have fully human-labelled dialogues (Q: Human, A: Human) collected in the dataset. We refer the reader to suppl. material for further information about the data collection.

Figure 4 shows the results where we present three combinations of Questioner$^Q$ and Answerer$^A$: ChatGPT$^Q$ & Human$^A$ (blue dot line), ChatGPT$^Q$ & BLIP2$^A$ (blue solid line) and the reference of Human$^Q$ & Human$^A$ (red dot line). We observe that while all experiments perform similarly, up to 5 dialog rounds, beyond that point, Human$^Q$ & Human$^A$ (red dot line) outperforms both ChatGPT alternatives. Considering the previous experiment in Figure 3, showing the advantage of ChatGPT over Human as questioner, we observe that Human generated answers are of better quality than BLIP2 (in terms of final retrieval results). This advantage boosts the Human full loop to become more effective than ChatGPT. The results imply a small domain gap between BLIP2 and Human answerer (but with similar trend), justifying the usage of BLIP2 in our evaluations. Furthermore, we observe that in the real use-case of ChatGPT$^Q$ & Human$^A$ the model reaches a high performance of 81% Hit rate using only 5 dialog rounds. Similar to previous results, we observe a saturation in the marginal benefit of the last dialogues rounds.

Next we show some qualitative examples. More examples as well as dialogues *e.g.* Human vs. BLIP2 answers can be found in the suppl. material. Figure 5 describes a search for *a traffic light*, where the answerer is BLIP2. At round 0 (search by the caption), the target image is ranked as $1,149$ due to existence of many traffic lights in the corpus and also erroneous candidates. The results show the rapid decrease in the rank reaching at the top of the candidates after only 3 dialog rounds. Note that

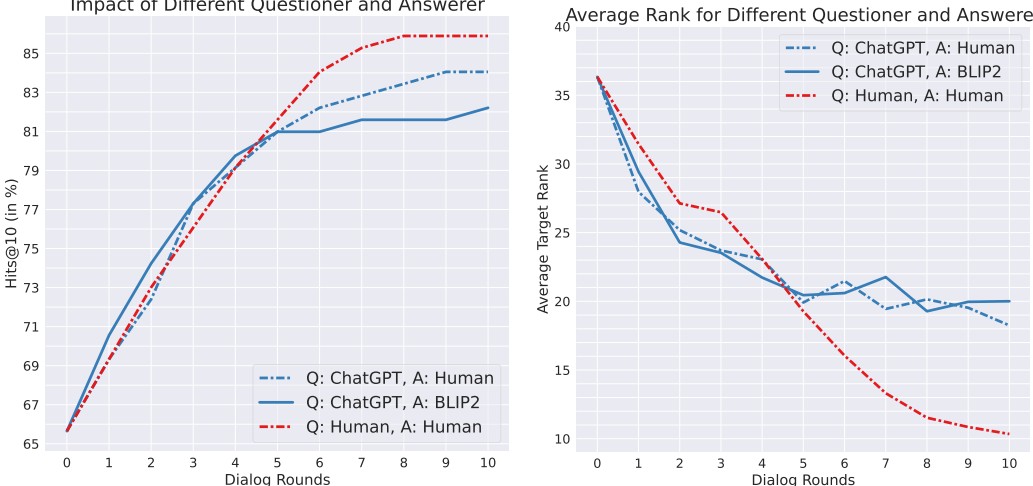

(a) Questioners Hits@10 performance (higher is better).   (b) Target image average rank (lower is better).

Figure 4: Human-AI. 1. We want to show performance on real scenario. 2. Verify the validation method BLIP2 vs. Human answerer 3. Human-Human is the best. BLIP2 answers are lower quality according to (a) We conducted this experiment to evaluate the difference, but still BLIP2 still presents a reasonable performance although inferior to human

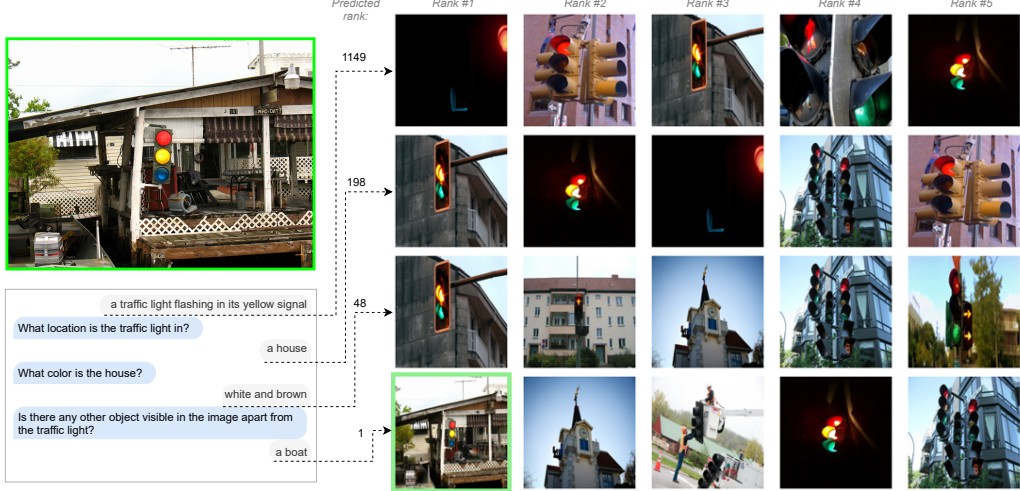

Figure 5: ChatIR example: a dialog is conducted between ChatGPT and a user imitator (BLIP2) on the ground truth image (green frame). Top-5 retrievals are presented after each dialog round.

since the first round, when the location is specified as "a house", the retrieved images are more likely traffic lights with a house in the background. Figure 6 shows another case with search for a specific *train*. This example is drawn from our test case against Human answerer. Note how the train at top of the list changes from black to *green and blue* after a question about the color is asked (second round), boosting the rank from 22 to 2. The last round fine-tunes the top-2 results by distinguishing between parking forms of *on track* vs *on platform*.

We conducted further analysis of using our pipeline for generating additional data, wherein the questioner and answerer collaborated to generate dialogues on more images. The inclusion of these dialogues and their corresponding images in the training set resulted in an improvement in the Average Rank metric, but did not provide any benefit to the specific retrieval measure of Hit@10. Due to lack of space, we discuss these results in the suppl. material.

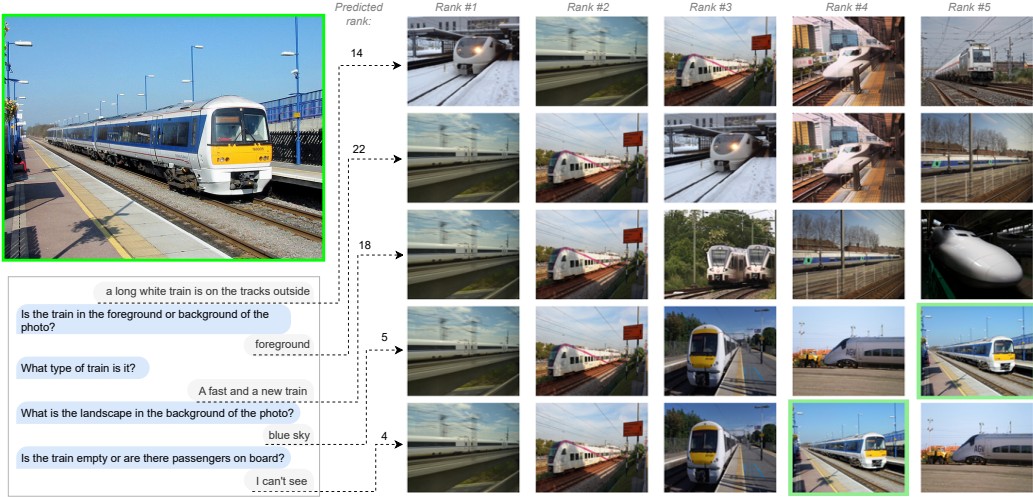

Figure 6: ChatIR example: a dialog is conducted between ChatGPT and Human on the ground truth image (green frame). Top-5 retrievals are presented after each dialog round.

## 5 Ablation Study

In this section we conduct an ablation and examine different strategies for training the Image Retriever model $F$. We further examine a few Questioner ($G$) baselines and discuss our evaluation protocol.

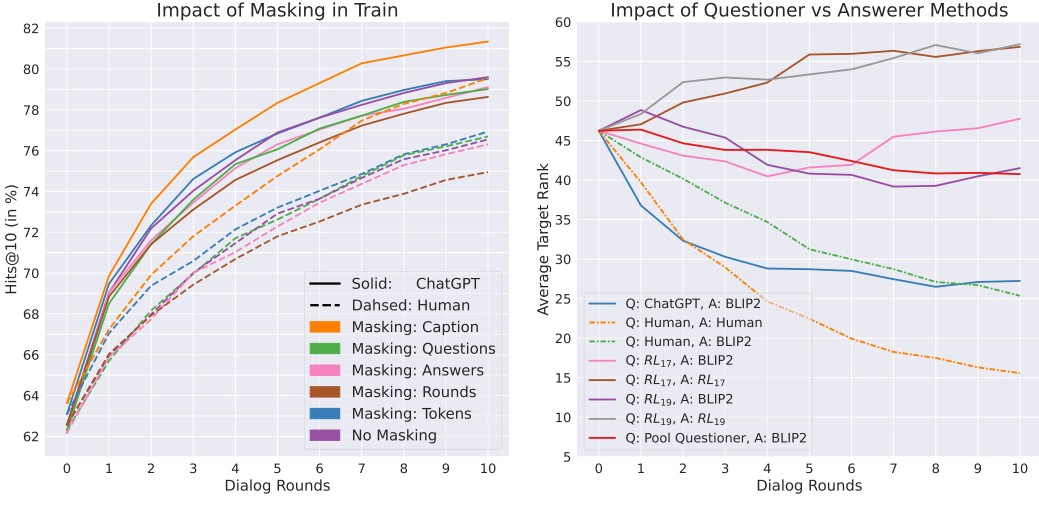

(a) Comparison of masking strategies.

(b) Rank performance for different Q&A models.

Figure 7: **Left:** Retrieval performance of image retriever models ($F$) trained with different masking strategies. Results are reported for two different answerers $G$. **Right:** Impact of different Questioner and Answerer models on the average target ranking, as the dialog progresses (lower is better). $RL_{17}$ [9] and $RL_{19}$ [34] represent previous RL methods.

**Masking strategy:** In this experiment we examine five different masking strategies for training of the Image Retriever model ($F$). Figure 7a shows evaluations of the resulting $F$ models using two different questioners $G$: ChatGPT (solid lines) and Human (dashed lines). As a baseline we train $F$ using dialog sequences (concatenated as described in Section 3), without masking any parts of the dialog. Next, we randomly mask different components of the training dialogues: captions, questions, answers, entire Q&A rounds, or individual tokens. In each strategy, we randomly select 20% of the components of a certain type for masking. The results in Figure 7a show that among these strategies, masking the captions improves the performance by $2 - 3\%$ regardless of the questioner type. By

hiding the image caption during training, F is forced to pay more attention to the rest of the dialogue in order to extract information about the target image. Thus, $F$ is able to learn even from training examples where retrieval succeeds based on the caption alone.

**Question Answering methods:** In Figure 7b, we examine different combinations of questioner and answerer in terms of Average Target Rank (ATR) and observe some interesting trends. We observe that Human and ChatGPT questioners are the only cases that improve along the dialog. As expected, Human answerer generates higher quality answers resulting in lower ATR. For this comparison we also consider a "pool" questioner, *i.e.*, a classifier that selects a question from a closed pool of 40K questions, as well as dialogues generated by two previous RL-based methods [9, 34].

We first examine the impact of using BLIP2 as a substitute to a human answerer, in our evaluation protocol. The same set of human generated questions (from the VisDial validation subset) is used to compare the changes in average target rank when using BLIP2 (green dash-dot) or a human (orange dash-dot) answerer. While BLIP2 is less accurate than Human, both follow the same trend. This justifies our use of BLIP2 to compare between different questioners at scale (Figure 3). In fact, the performance measured using BLIP2 can be considered as an underestimation of the true retrieval performance. Figure 8 shows an example with answers of BLIP2 and a human (to human questions).

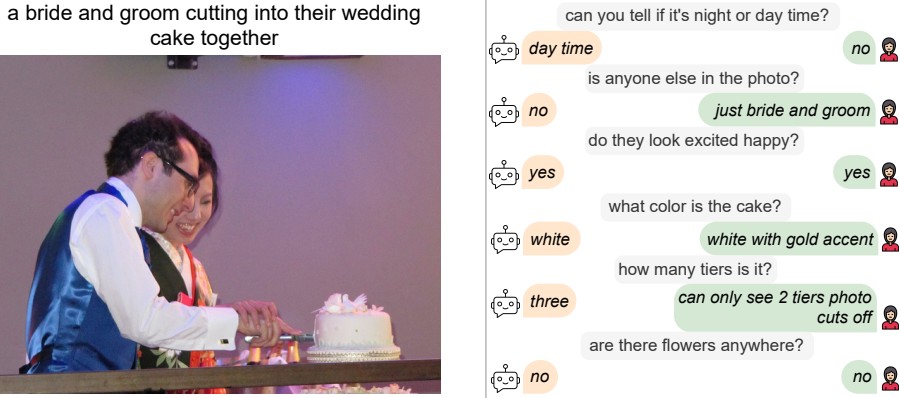

Figure 8: Dialog example with two different answerers: BLIP2 (left) and Human (right).

Next, we examine the rank performance of dialogues generated by two previous RL-based methods, $RL_{17}$ [9], and $RL_{19}$ [34]. We compare dialogues entirely generated by each of the two methods, as well as dialogues where the questions of $RL_{19}$ are answered by BLIP2. We observe that the RL-based answerer (dubbed A-bot in [34]) performs poorly, since replacing it with BLIP2 significantly improves the results (dashed gray and brown lines vs. dashed pink and magenta). However, even with the BLIP2 answerer, the RL methods still struggle with improving the average retrieval rank as the dialog progresses, and the rank remains nearly constant, similarly to the pool questioner model.

## 6   Summary and Discussion

Since conversation is a natural means of human information inquiry, framing the visual search process within a dialog is expected to make the search process more natural, in terms of query entry and interaction to locate relevant content. In this paper, we proposed ChatIR for image retrieval, a model that chats with the user by asking questions regarding a search for images, being capable of processing the emerged dialog (questions and answers) into improved retrieval results. We showed through extensive experiments that using foundation models we are able to reach a performance level nearly as good as a human questioner. Our analysis yields some interesting insights and results: *e.g.*, a failure cause in many alternatives appear to be the inability in continuously generating new genuine questions. Yet, some limitations still exist, *e.g.*, our current concept uses a questioner that relies solely on the dialog as an input, to generate the follow-up question. An optimal questioner however, may further consider the retrieved results or a set of candidates in order to extract the most distinguishing attribute for narrowing down the options. We believe that our framework and benchmark will allow further study of the demanding application of chat-based image retrieval, as a tool for improving retrieval results along with continuous human interactions.

**Acknowledgments:** This work was supported in part by the Israel Science Foundation (grants 2492/20 and 3611/21). We would like to thank Elior Cohen, Oded Ben Noon and Eli Groisman for their assistance in creating the web interface. We also thank Or Kedar for his valuable insights.

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
