# Chatting Makes Perfect: Chat-based Image Retrieval

## Supplementary Material

In Appendix A, we start by showing more qualitative results of chats and their retrieval results, and BLIP2 chats compared to a human answerer. Next, in Appendix B, we present the few shot instructional prompts that were used by different LLMs for generating follow-up questions. We report details about our real ChatIR experiment in Appendix C, where we conduct chats with human in the loop (as answerers). In Appendix D, we elaborate on training the image retriever $F$ with more analysis. We further examine the influence of *Visual Dialog Generation* for training $F$ on extra data, in Appendix E. In Appendix F, we briefly discuss the types of question and answers appear in dialogues. Finally, we report statistics of question repetitions of different Questioners, in Appendix G, that we found to be related with questioner performance.

## A    Qualitative Results

Here we present more examples for chats and retrievals along the dialog. Particularly, in Figure 1, we show an example of imperfectness in the image retriever model $F$. The "good" question regrading the color raised by ChatIR, significantly affects the target's image rank bringing to 7th place. Note that although in the first place, there is no pink umbrella, we believe that the model is mistakenly relying on the pink color of the human shirt in the picture. Eventually, after two more questions, asking about the "ethnicity" and location, the model is able to find the desired image, ranked in the 2nd place. Another example in Figure 2 describes two trains, searched by the text "A train that is parked next to another train". It shows a monotonic decrease in the rank up to the 1st place, after only 3 rounds, when asked about the train type and color and the location of parking.

Figure 3 demonstrates a case where the description "a small and dirty kitchen with pots and food everywhere" is ambiguous, subjective to the viewer and may match many images in the corpus. However, the chat allows the user to enrich the query using only a few Q&A rounds, narrowing down the number of relevant candidates.

In Figure 4 we show an example of a dialog between ChatIR and a human. In this case, the initial description is general and in practice insufficient to pinpoint the single desired image, as many other images apply to the same description. These examples also known in literature as "false negatives" [3] depend also on the content of the searched dataset (how similar are some images to others).

Next, we present in Figure 5 two examples with a comparison between BLIP2 and Human answerer to the same questions. We observe that Human answers tend to be longer (probably more complete), often adding relevant information voluntarily. To reinforce this observation we calculate the average number of unique tokens per answer of Human and BLIP2 (to the same questions), showing 8.24 tokens for BLIP2 versus 25.29 tokens for human. This observation matches also the results reported in the paper showing improved retrieval performance with Human answerers.

## B    Few-shot Questioners

In Section 4 of the main paper we discuss different questioners based on LLM few-shot. Here we provide a specific example for our few-shot prompt, that generates a follow-up question in a dialog. Given the following partial dialog of 2 rounds:

"*Caption: a group of people standing on a snowy slope*
*Question: Are there any trees visible in the background of the image?*
*Answer: no*
*Question: How many people are in the group?*
*Answer: four* "

We prompt the LLM model with the task description (green) attached with a train example (blue), as the following:

"*Ask a new question in the following dialog, assume that the questions are designed to help us retrieve this image from a large collection of images:*
*Caption: 2 full grown zebras standing by a brick building with a steel door*

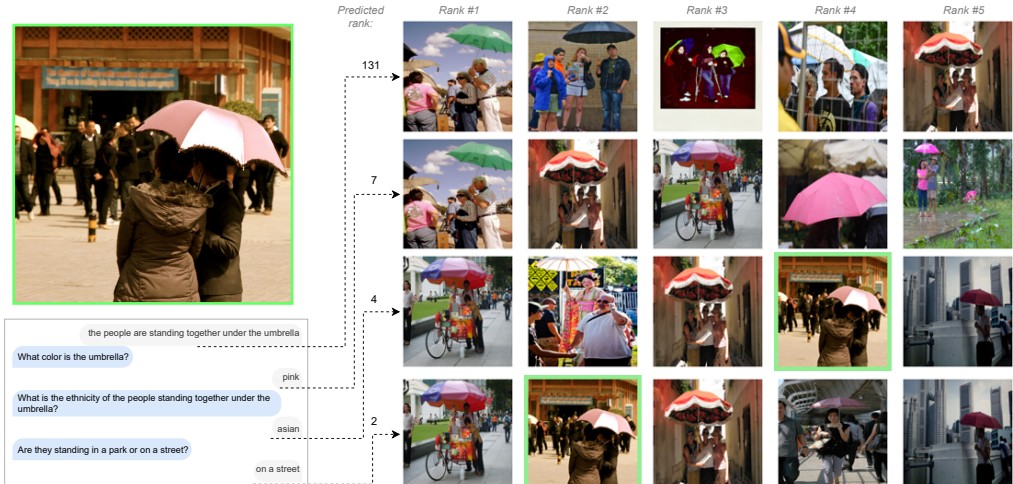

Figure 1: ChatIR example: a dialog is conducted between ChatGPT and a user imitator (BLIP2) on the ground truth image (green frame). Top-5 retrievals are presented after each dialog round.

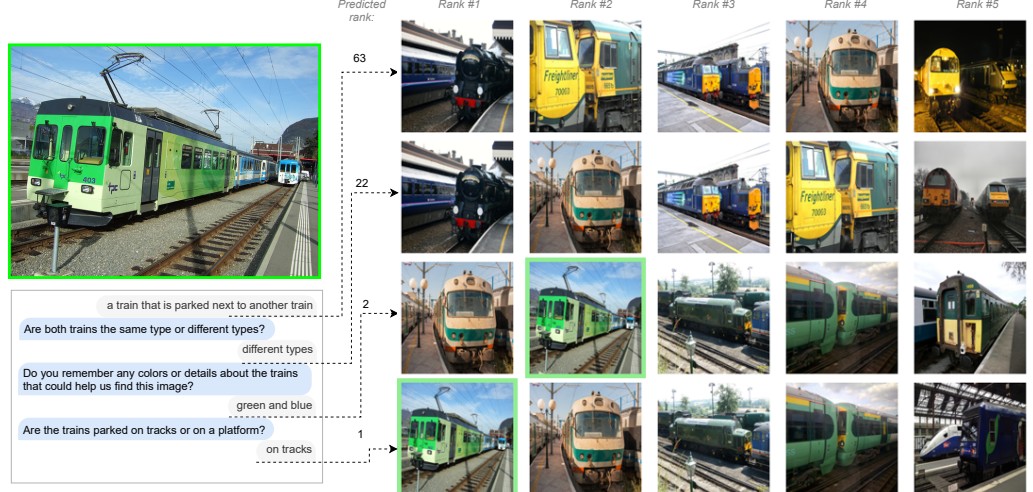

Figure 2: ChatIR example: a dialog is conducted between ChatGPT and a user imitator (BLIP2) on the ground truth image (green frame). Top-5 retrievals are presented after each dialog round.

*Question: is this picture in color?*
*Answer: yes*
*Question: do you see people?*
*Answer: no*
*Question: are the animals in a pen?*

 *Caption: a group of people standing on a snowy slope*
*Question: Are there any trees visible in the background of the image?*
*Answer: no*
*Question: How many people are in the group?*
*Answer: four*
 *Question:* "

Where the blue dialog is a train example, consist of two rounds of Q&A + the human-labelled follow up question. We add an empty "*Question:*" line following the partial dialog (orange) allowing the LLM to complete.

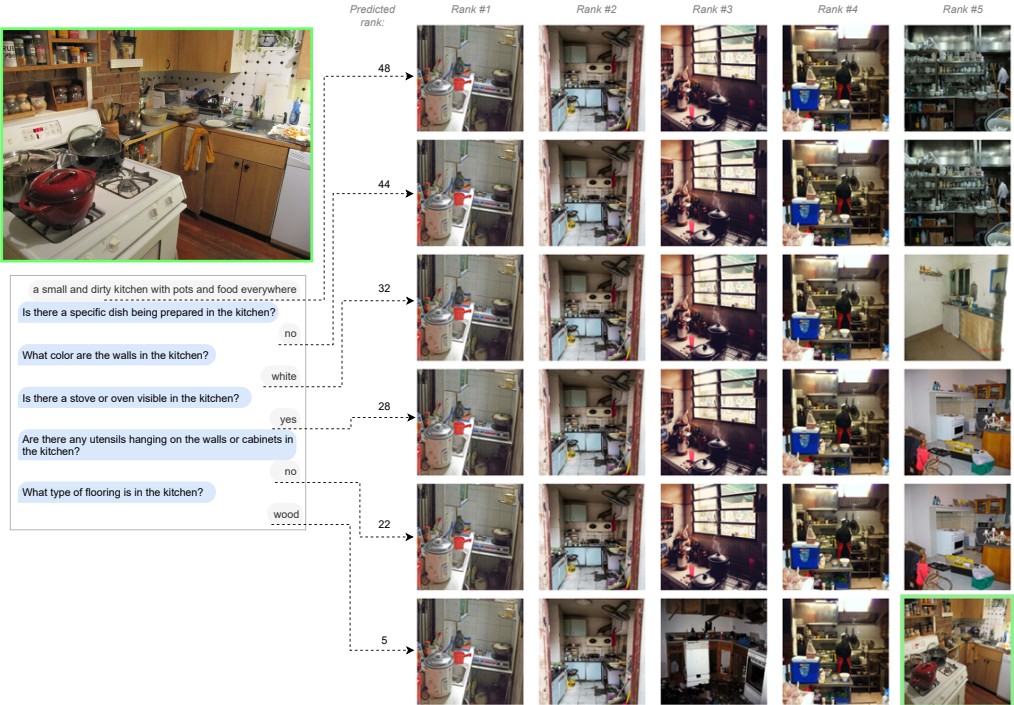

Figure 3: ChatIR example: a dialog is conducted between ChatGPT and a user imitator (BLIP2) on the ground truth image (green frame). Top-5 retrievals are presented after each dialog round.

## C  Real ChatIR Experiment

Here we provide details about the study where we harvest data from Human answerers (discussed in Section 4.2, the main paper). We follow a similar protocol the one used in VisDial [1]. In this experiment users are provided with a specific image and they are asked to answer questions asked regarding that image. Figure 6 shows a screen shot of the designed web interface that we used to collect the answers. We collected 163 different dialogues that are summarized to a total of 1630 Q&A rounds. Participant age range is between 17 and 60, summarized to a total of 93 different participants, presented in Figure 7. Our participants were composed of students and colleagues from Israel and the United States. Our participants were instructed by the following description:

> *"Dear All,*
> *We would like to invite you to take part in a brief user study that we are conducting for our research on dialogs on images. There is a chatbot behind the scene that generates questions regarding an image, based only on the initial image caption. We need your answers to allow the chatbot obtain more information regarding the content of the image. As part of the study, a chatbot will pose you 10 short questions about an image, and we kindly request that you provide your responses. Your participation in this study would be greatly appreciated. If you are willing to help by participating, please click on the following link:* **\*\*LINK\*\***
> *Thank you for your time and consideration. "*

## D  Image Retriever Training

In this section we test the consistency of our image retriever $F$ training over different seeds. Figure 8 presents the results in terms Recall@K and Average Target Rank when tested on human Q&As. To this end, we trained our $F$ with the caption-masking strategy (see Section 4 in the main paper) five

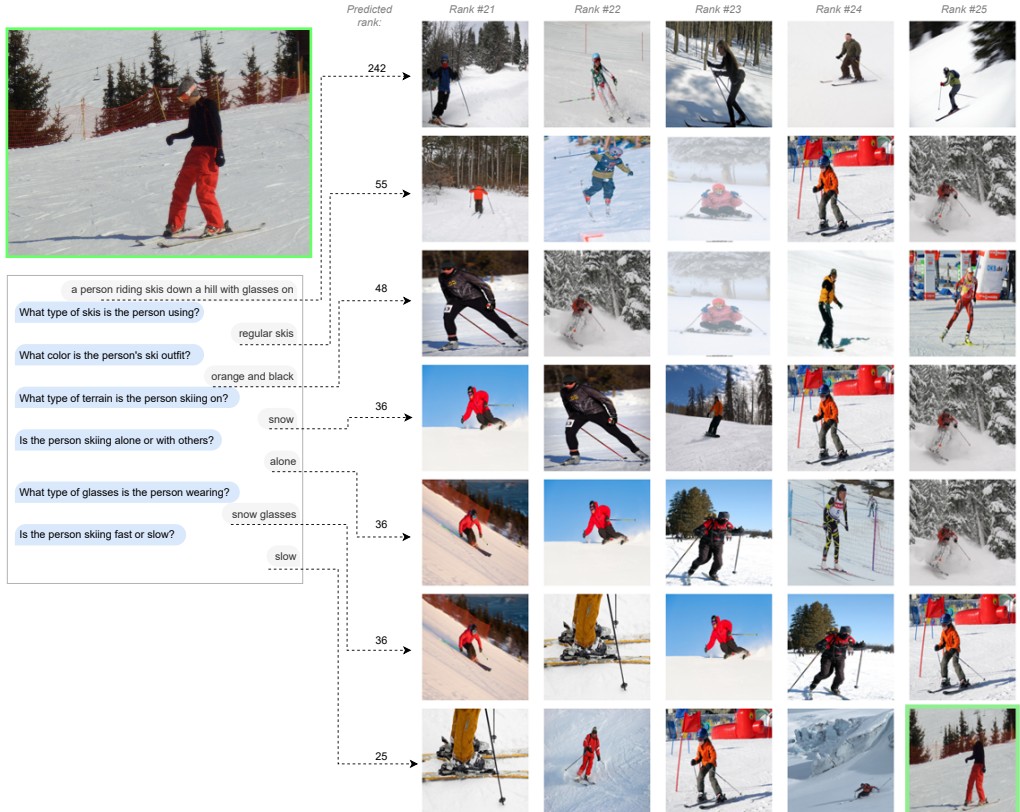

Figure 4: ChatIR example: a dialog is conducted between ChatGPT and a human on the ground truth image (green frame). Rank 21-25 retrievals are presented after each dialog round.

times with random initial seeds. As observed from the standard error bars, our results are robust over initial seeds.

# E    Data Generation

We examine the usage of *Visual Dialog Generation* task to generate additional dialogues for training our image retriever $F$. To this end, we use ChatGPT (as questioner) and BLIP2 (as answerer) to generate new dialogues on either the existing or new images. We examine the following two different settings for generating new training data:

1. Creating alternative synthetic dialogues for each image from existing training set, namely dialogue augmentation. For this setting, we train $F$ on the emerged doubled training set (consisting of two dialogues per image).

2. We generate dialogues on $73K$ *new images* from COCO [4] unlabeled set, that were excluded from the original VisDial training and validation sets. Next, we again train $F$ on this emerged dataset that includes 68% more images than the original set.

Figure 9 shows results of the first setting (dashed) and the second's (dotted). Here, we observe different trends according to two different measures, Hits@10 and ATR. In Hits@10 (Figure 9a), where we stop using further dialogues when the image reaches top-k rank, we see similar or moderate reduction in performance. However, for ATR (Figure 9b) encapsulating the average performance through all 10 rounds, we see improvement (lower ATR) after 2-dialogue rounds, as we augment dialogues with our pipeline (ChatIR), while training with additional new images harmed the model.

We draw two main conclusions from these tests: (1) The results show that the efficacy of dialogue augmentation depends on the measure and the use-case. (2) We believe that the cause for reduced

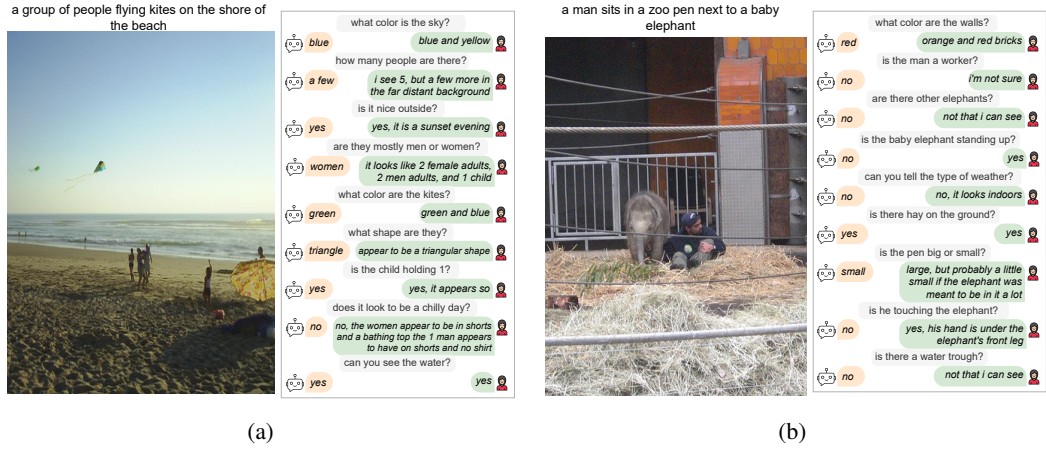

Figure 5: Dialog examples with human questions answered by two different answerers: Human (green right) and BLIP2 (orange left). Note that human answers tend to be longer, often with voluntarily added information.

Figure 6: Screen shot of the web-interface used for collecting human answers.

performance in new-images with dialogues is in the poor captions (compared to manual captions), that were generated by BLIP2. We observe this effect in Figure 9b by the decreased performance at zero-round (text-to-image retrieval-dotted lines) with 4% drop in ATR, resulting a lag in performance with respect to "vanilla" setting. The difference between the Hits@10 and ATR can be explained by the fact that when measuring ATR, top-k retrievals considered successful at certain point continue to change along further dialogues, and can descend in the rank. The improvement achieved by dialogue augmentation (dashed lines) decreases this harmful effect. We leave the further analysis of this feature for future study.

## F Types of Questions and Answers in Dialogues

Since the end goal of ChatIR is to retrieve the image, we expect the questioner to generate questions that, when answered, improve the retrieval results (which is the instruction we give to ChatGPT, see Appendix B). In Figures 10 and 11 we visualize the word occurrences in questions and answers from

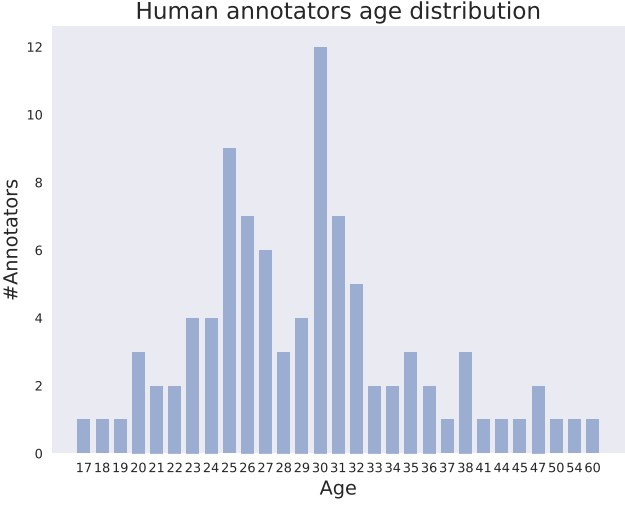

Figure 7: Distribution of human annotator ages.

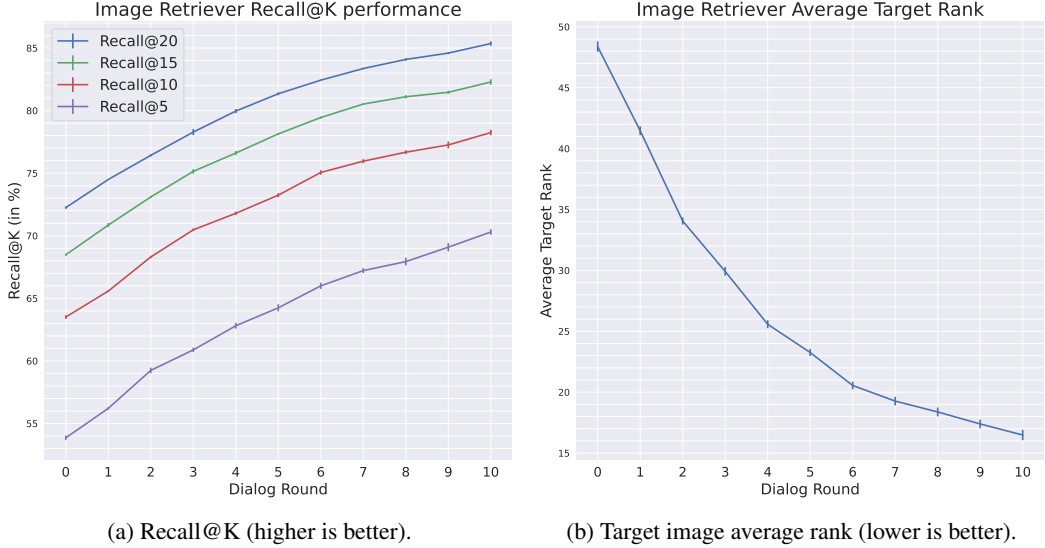

(a) Recall@K (higher is better).

(b) Target image average rank (lower is better).

Figure 8: Image Retriever ($F$) performance with STD error bars, on five different experiments of training our image retriever $F$. (a): Recall@K (left), with four different values of $K$. (b) Average Target Rank. The small size of the error bars indicate consistency over different seeds.

both human and LLM-generated dialogues. According to our analysis and visual inspection of the dialogues, these questions and answers typically relate to color, location, time of day, size or number of certain objects, and more (see also Figures 5 and 6 in the paper, and Figures 1 to 5 for both LLM and human questioners).

## G   Question Repetitions

In Section 4.1 we discuss repetitions in questions, asked by different questioner models. To this end we used the simple statistics of counting the average number of unique tokens in 10 dialogue rounds. Figure 12 presents these statistics showing that "bad" questioner models are associated with more repetitions (lower average unique tokens) in the questions, as expected.

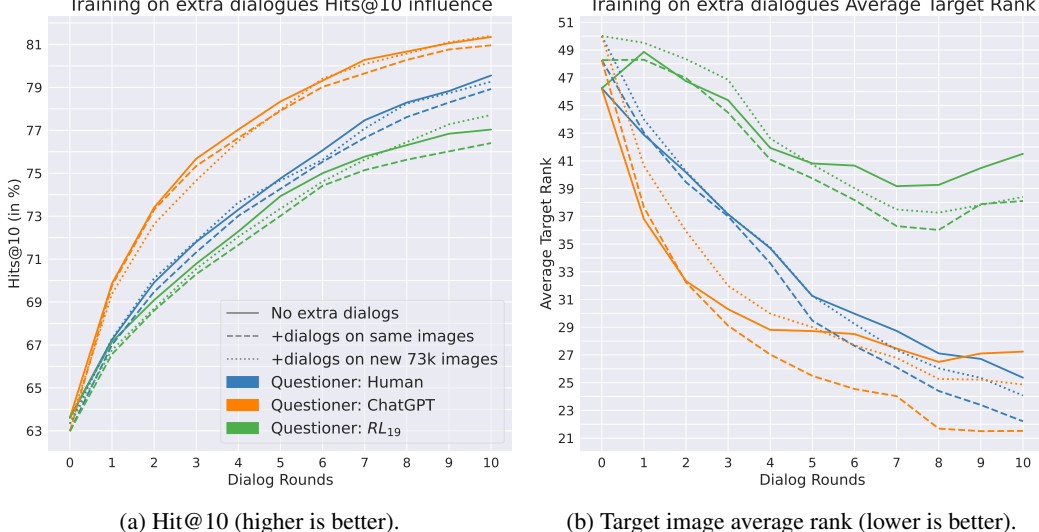

(a) Hit@10 (higher is better).  (b) Target image average rank (lower is better).

Figure 9: Influence of training $F$ on extra synthetic dialogues. We compare between "vanilla" training on the VisDial dataset (solid lines) and two different train-set extension methods: (1) Training on additional augmented dialogues (dashed line) (2) Additional augmented dialogues on new images (dotted lines). We show results on different Questioner models. Overall the results show no-improvement in Hits@10 measure but with noticeable gain in ATR measure.

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

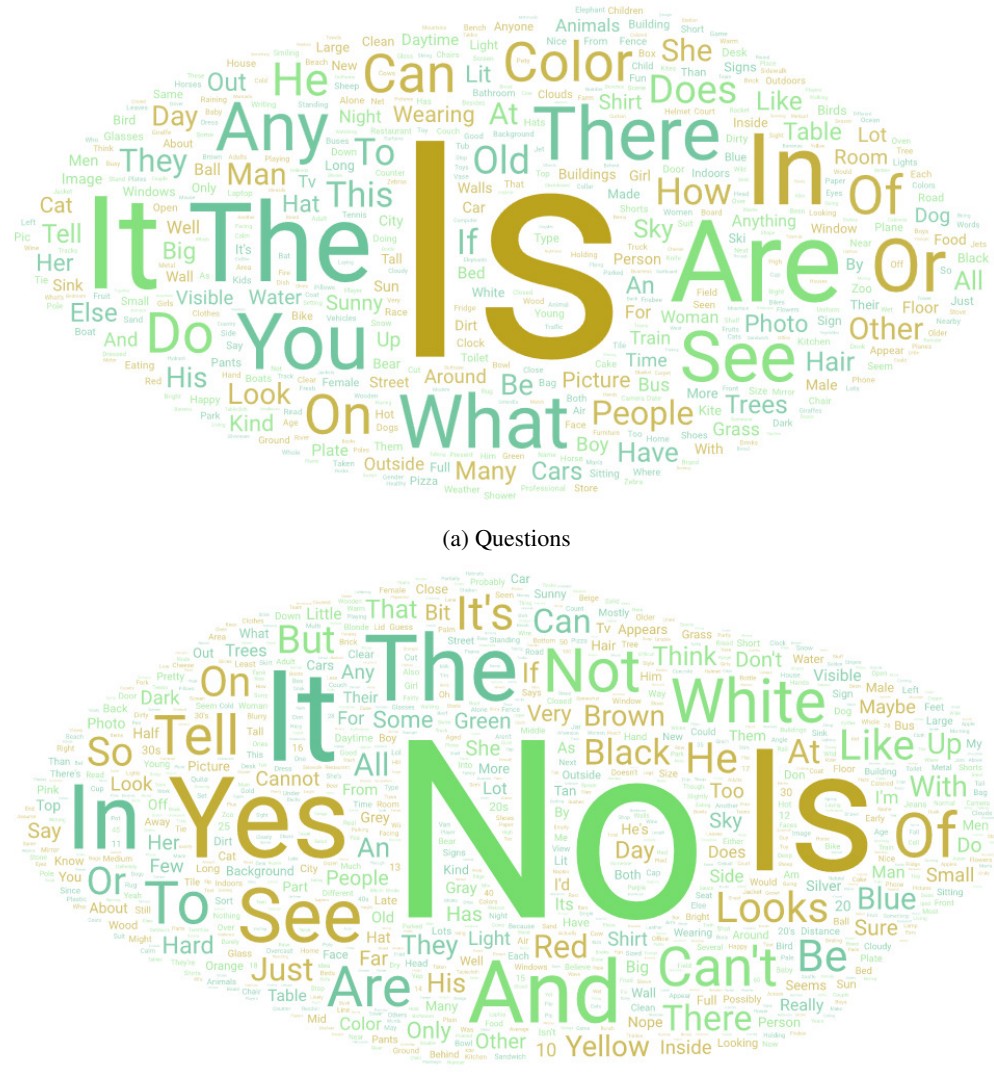

(a) Questions

(b) Answers

Figure 10: Top 500 most frequently occurring words in human questions (a) and answers (b), from the VisDial test set (larger size means more frequent).

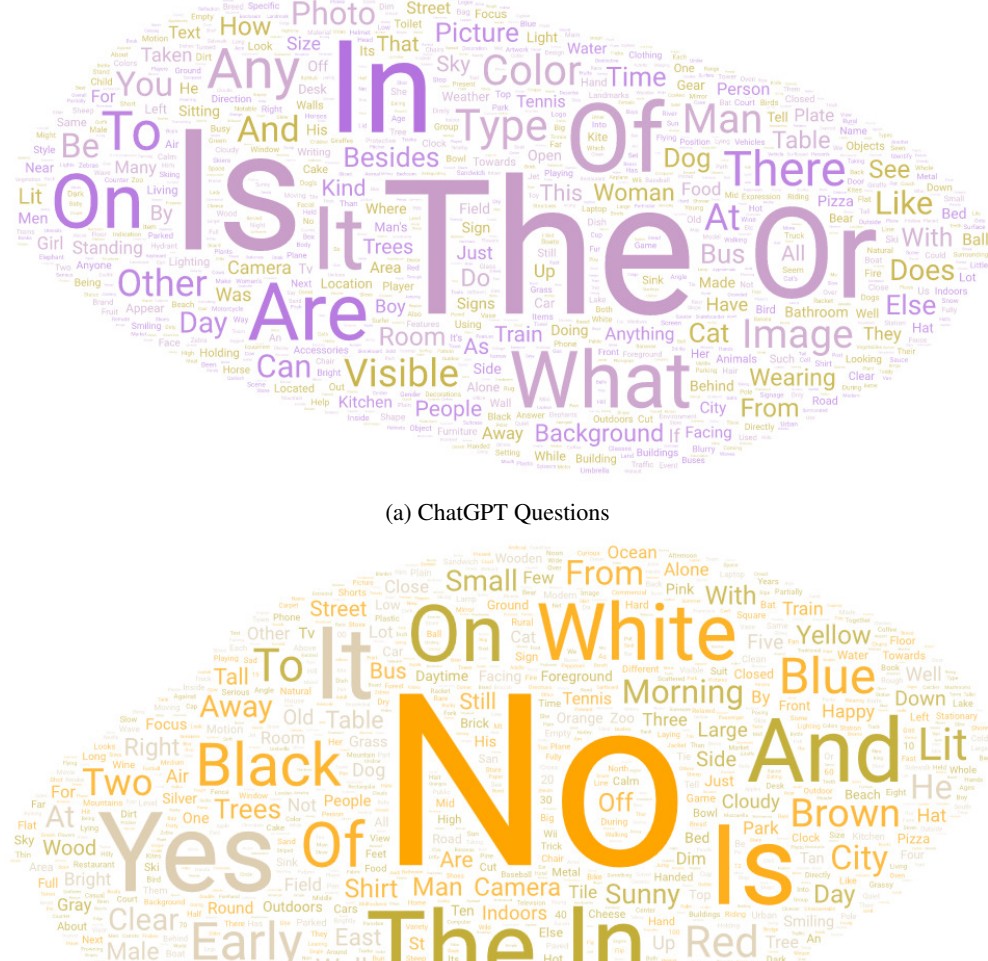

(a) ChatGPT Questions

(b) BLIP2 Answers

Figure 11: Top 500 most frequently occurring words in questions generated by ChatGPT (a) and answers generated by BLIP2 (b), on the VisDial test set (larger size means more frequent).

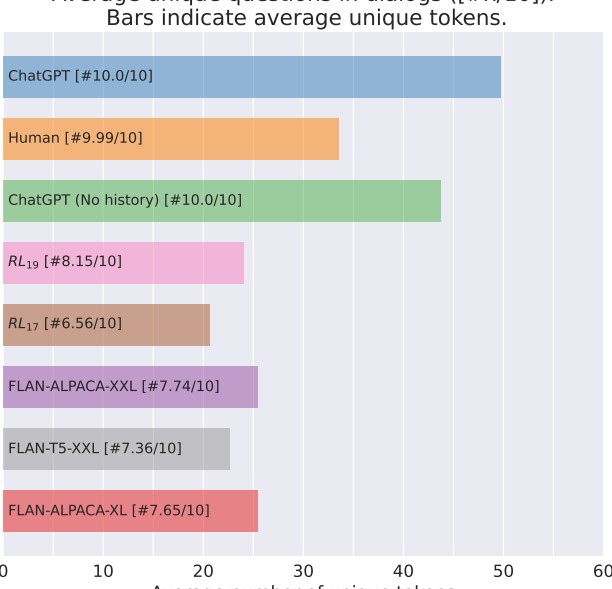

Figure 12: Indications for question repetitions in dialogues, by different questioner methods. $RL_{17}$ and $RL_{19}$ stands for Reinforcement Learning methods [2] and [5], respectively. "Bad" questioner models are associated with more redundancy (lower average unique tokens) in the questions, as expected (see main paper).