# OpenReview forum: "Chatting Makes Perfect: Chat-based Image Retrieval"
_NeurIPS.cc/2023/Conference — NeurIPS 2023 poster_

### Official Review · Reviewer_L59s · 2023-07-02

**Soundness:** 3 good
**Presentation:** 3 good
**Contribution:** 4 excellent
**Rating:** 5
**Confidence:** 4

**Summary:**

The paper tackles the problem of chat-based image retrieval. The goal is that the system, being chat-based and powered by a LLM, engages in a conversation with the user to elicit information, in addition to an initial query, in order to clarify the user’s search intent by asking follow up questions. These questions form a dialog with the user in order to retrieve the desired image from a large corpus. The authors also suggest an evaluation protocols suitable for continual progress and assessment of questioners using a Visual Dialog model in place of a human and test their framework on real human interactions which involves collecting answers from users and further evaluate the method against strong baselines generated from prior art.

**Strengths:**

The paper tackles an interesting problem and proposes an interesting setup as opposed to the text-image retrieval task. If all data and the protocol is made available, I think this can be very valuable for the research community.

**Weaknesses:**

I really like the overall goal of the paper and I think it's an interesting direction. However, I have few concerns in terms of clarity (please see questions below) that I think need to be further detailed in the paper. Also, what I think it's lacking the current version is a comparison with classical text-image retrieval methods. If I understood correctly, the method can be applied to any dataset, so I think that applying it to some common benchmarks for text-image retrieval and showing how the dialog rounds affect the results can further validate this approach.

**Questions:**

1. Are there other methods than BLIP2 that can be used to answer the questions? Based on the results from Fig 3, up to a certain dialog round of 4-5 there is no difference between human answers and BLIP2. Do the authors have any insights on what happens beyond that point?

2. Did the humans have access to the image when answering the questions?

3. What data do you use for evaluation since it's not clear? VisDial?

4. Will all the data and all the evaluation protocol be made available online?

5. Can you elaborate a bit on the Unanswered setup? The model has available the caption + several questions but no answer when making the retrieval?

**Limitations:**

Various limitations are discussed throughout the paper and there the societal impact is somehow discussed, but briefly.

---

> ### Author Rebuttal · Authors · 2023-08-09
>
> Dear Reviewer **L59s**, thank you for your insightful feedback!
>
> **Lack of comparison with classical text-image retrieval methods:**
>
> Please see the global part of the rebuttal and the figures in the accompanying pdf, where we address this concern.
>
> ### Answers to questions: ###
>
> 1.  **“Are there other methods than BLIP2 that can be used to answer the questions?”**:  Theoretically, any generative Visual-Dialog model might serve as an answerer. We chose the SoTA model for this task (BLIP2).
> **“Fig 3, up to a certain dialog round of 4-5 there is no difference between human answers and BLIP2. Do the authors have any insights on what happens beyond that point?”**:  Our analysis shows that human answers are longer (see analysis in lines 29-34 in suppl. material) implying that humans tend to volunteer additional information, while BLIP2 provides short and concise answers (e.g. Fig 7, and suppl. Fig 5). Another source of this discrepancy might be the less accurate and less elaborate BLIP2 answers at long dialogues.
>
> 2.  **“Did the humans have access to the image when answering the questions?”**: Yes, in all experiments that involved ground truth human answers, users had access to the image.
>
> 3.  **“What data do you use for evaluation since it's not clear? VisDial?”**: Yes, our evaluation was primarily based on different measures for the image retrieval task. We use the VisDial dataset (lines 55-59), where each image is associated with a dialog. Thanks to VisDial we are able to test ChatIR in many different scenarios, such as combining a human questioner and a machine or human answerer. We test these different dialog options and measure the retrieval capabilities of images from the pool (50k). As mentioned earlier, for this rebuttal we also use COCO and Flickr30k with synthetic dialogues .
>
> 4.  **“Will all the data and all the evaluation protocol be made available online?”**: Yes, the entire data and protocols will be made available upon acceptance.
> 5. **“Can you elaborate a bit on the Unanswered setup?”**: Following the description in lines 194-200, in this setting the *questioner* (ChatGPT) is provided with the caption only. It is asked to generate at once 10 different questions based solely on the caption (without seeing any answers). Thus, although the retrieval is conducted using the full dialogues (questions and answers), in this setting the answers have no effect on question generation in the dialog. We will clarify this in the revision. This experiment also shows the answers’s influence on the question generation, namely, questions conditioned on the prior dialogue history are more effective for retrieval.

---

> > ### Comment · Reviewer_L59s · 2023-08-17
> > **Rebuttal**
> >
> > I have acknowledge that I read the rebuttal.
> >
> > The authors have addressed some of my concerns. I think that the setup is interesting, hence I am raising my score to Borderline Accept. I still think that some more comparisons with other methods, especially a comparison with classical retrieval methods would be beneficial in better understanding the new retrieval setup.

---

> > > ### Author Response · Authors · 2023-08-17
> > >
> > > Thank you for your response. We compared our method to CLIP and BLIP, which are considered SoTA with respect to all previous methods. However, we would be happy to compare with more methods that seem to be relevant. We welcome any suggestions as to which classical methods to compare with.

---

> > > > ### Comment · Reviewer_L59s · 2023-08-17
> > > > **Rebuttal**
> > > >
> > > > First off all, to clarify a bit, I am not saying that this is something that needs to be added in the current version, but I think it’s something of interest for future work.
> > > >
> > > > One of my curiosities about the chat based retrieval is how it compares to the regular task of text video retrieval. So, more exactly (though I am sure there are some complexities in achieving this) is to take a text video retrieval benchmark such as MSRVTT, ActivityNet, etc and use it for chat based video retrieval along with a Large Language Model to generate the chat part. I am curious how these additional question would affect the performance as opposed to the current SoTA for that particular dataset. One example of a method can be [1] (and you can find more methods in the SoTA comparison), but my curiosity is more about a comparison between chat-based retrieval vs regular retrieval.
> > > >
> > > > Again, this is something of a high level suggestion that I don't think is needed for this submission, but it's something that I think might be worth pursuing in future works.
> > > >
> > > > [1] Gorti, Satya Krishna, et al. "X-pool: Cross-modal language-video attention for text-video retrieval." Proceedings of the IEEE/CVF conference on computer vision and pattern recognition. 2022.

---

> > > > > ### Author Response · Authors · 2023-08-20
> > > > >
> > > > > This is indeed an interesting direction for future work, as video retrieval can be an extension of our work. This will require constructing a new dataset of videos-chats, with a video Q&A model and probably also temporal action detection. We are happy that this paper has raised new ideas such as this one. Our intention is also to stimulate/inspire the community about such options.

---

### Official Review · Reviewer_34K9 · 2023-07-03

**Soundness:** 2 fair
**Presentation:** 3 good
**Contribution:** 2 fair
**Rating:** 4
**Confidence:** 4

**Summary:**

In this paper, the authors present a dialog-based image retrieval system and show strong performance against baseline models.

**Strengths:**

1. The author's attempt to augment the image retrieval process with dialogue is interesting and largely under-explored.

2. The paper is well-written and easy to read.


**Weaknesses:**

1. The authors are largely missing out on what type of dialogue one needs to have (given the caption) such that it helps to better visualize the caption. Why one even needs to do that? Is it because the captions are not useful/detailed?
2. The experiments are solely designed around the availability of instruction-tuned LLMs (for question generation), BLIP2 (for answering), and the VisDial dataset. Once you answer my previous comment, please justify why VisDial (alone) is enough for this experiment. Are instruction-tuned LLMs generating the type of questions you expected?
3. The paper is not well motivated. Specifically, the need to have a dialogue for image retrieval. Though the results are encouraging the reason why we are doing these experiments in the first place is not clear. I request the authors to give some more thought to this.

**Questions:**

Check my comments for weaknesses.

**Limitations:**

It is not clear how this works could be expanded beyond the current setup majorly because it is not very clear what type of dialogue one needs to have with the system.

---

> ### Author Rebuttal · Authors · 2023-08-09
>
> Dear Reviewer **34K9**, thank you for your insightful feedback!
>
> 1: **Regarding the motivation and “What type of dialogue one needs to have (given the caption) …”**:
> The motivation is that, typically, a short query is not sufficient to retrieve the correct images from a large corpus, certainly not in a single attempt, as we discussed in lines 27-30 and 35-39. Commonly, people search an image with a short description, that might fail to fully convey the search intent, or sometimes it may result in many images that comply with the same description (please see Fig 4,5 and examples in suppl. material). Our idea is to engage in a conversation with the user in order to elicit additional information, and clarify the user’s search intent (lines 35-39). Gradually eliciting and accumulating the information from the user and being able to process it in a unified way is the essence of our ChatIR.
>
> **Type of questions:** Since the end goal of ChatIR is to retrieve the image, we expect the questioner to generate questions that when answered, improves the retrieval results (which is the instruction we give to ChatGPT). These questions typically relate to color, location, time of day, number of certain objects and more (see examples in Figs 4 and 5 in the main paper, and Fig. 1-5 in suppl. for LLM and human questioner).
>
> We show that such dialogues are able to significantly boost the retrieval performance (lines 173-176 and 208-210). Please also see the global part of the rebuttal and the figures in the accompanying pdf, where we discuss the advantages of our approach.
>
> 2.1: **“please justify why VisDial (alone) is enough for this experiment”**: Since our task requires data consisting of images paired with dialogues, we use the VisDial dataset that contains such annotations (although it was annotated for a different task), as we describe in lines 55-59. Thanks to VisDial we are able to test ChatIR in many different scenarios, such as combining a human questioner and a machine or human answerer. These results are presented in Fig 2 (paper) and Fig 8 (suppl. material).
>
> 2.2: **“Are instruction-tuned LLMs generating the type of questions you expected?”**: Yes. The end goal of ChatIR is to retrieve the image, thus we expect the questioner to generate a question that when answered, improves the retrieval results. This is indeed the case as our evaluations show (Fig 2, 3 and in suppl. material). More specifically, as we discuss in Sec 4.1, some Instruction-tuned LLMs generate the type of questions we expected (e.g. ChatGPT, see examples in Fig 4, 5 and Suppl.material). On the other hand, some others (e.g. FLAN-T5, FLAN-ALPACA) struggle with long context, which results in question repetitions (lines 224-231) resulting degraded performance (Fig 2).
>
> 3: **Motivation**
>
> Please see our response above

---

### Official Review · Reviewer_Z6DK · 2023-07-06

**Soundness:** 2 fair
**Presentation:** 3 good
**Contribution:** 2 fair
**Rating:** 4
**Confidence:** 3

**Summary:**

This study introduces ChatIR, a chat-based image retrieval system that engages in a conversation with the user to clarify their search intent and retrieve the desired image from a large corpus. The system leverages Large Language Models to generate follow-up questions to an initial image description and achieves a success rate of over 78% after 5 dialogue rounds, compared to 75% when questions are asked by humans and 64% for a single shot text-to-image retrieval.


**Strengths:**

The strength of this submission lies in its clear and compelling motivation, which focuses on utilizing chat interactions to refine search and enhance image retrieval. The authors effectively articulate the significance of this research direction, highlighting the potential of chat-based interactions to improve the accuracy and relevance of image search results. Additionally, the submission features a nice illustration that visually communicates the proposed approach, providing a clear representation of the underlying concept. This visual aid aids in understanding the methodology and reinforces the clarity of the paper.


**Weaknesses:**

One notable weakness of the submission is the lack of baseline comparison. While the paper introduces a novel conversation-based setup for image retrieval, it only reports results within this specific setup. The absence of results on the traditional single-hop text-to-image retrieval using the same dataset raises concerns about the necessity of introducing conversation into the retrieval process. Without a comparison to traditional single-hop methods, it is difficult to fully understand the advantages and potential improvements offered by the conversation-based approach (what if traditional methods already achieve comparable success rates?). Such comparisons would help address the question of why the system should be made more complex with conversation, and provide a stronger rationale for the proposed approach.


**Questions:**

(1) Missing reference for "common practice in image retrieval" in line 148.

(2) The readability of Section 4 is currently hindered by the complexity of the annotations, such as the format of "Q: XXX & A: YYY." To enhance readability, I recommend using a different font or formatting approach to simplify the annotations in the revised version. Simplifying the annotations will make the section more accessible and easier to follow for readers, ensuring a smoother comprehension of the methodology and findings.



**Limitations:**

The authors have included discussions of the limitations.

---

> ### Author Rebuttal · Authors · 2023-08-09
>
> Dear Reviewer **Z6DK**, thank you for your insightful feedback!
>
> **The absence of results on the traditional single-hop text-to-image retrieval:**
>
> Please see the global part of the rebuttal and the figures in the accompanying pdf, where we address this concern.
>
> Q1: Missing reference: Thank you. We will add it.
>
> Q2: Readability Q: XXX & A: YYY: We appreciate this suggestion and will change it in the revised version.

---

> > ### Comment · Reviewer_Z6DK · 2023-08-10
> >
> > Thanks for providing the rebuttal response.
> >
> > I acknowledge that I have read the response and the additional material in the provided PDF.

---

### Official Review · Reviewer_6Lfv · 2023-07-07

**Soundness:** 3 good
**Presentation:** 3 good
**Contribution:** 3 good
**Rating:** 5
**Confidence:** 4

**Summary:**

This paper proposes a chat-based image retrieval framework, which can clarify users’ search intent. Authors design a question generation model based on LLM to generate questions based on dialog history. After user answer the question, an image retriever which is a transformer model is trained to extract text embedding to search image. Authors also use a LLM to answer the questions taking the place of users for fast training and evaluation. Authors use existing dataset to evaluate the method.

**Strengths:**

+ The proposed framework is useful for clarifying user search intent, thus it’s practically valuable.
+ The proposed framework is well evaluated.
+ Authors use a existing dataset to avoid collecting a new dataset.

**Weaknesses:**

--- Components of the proposed framework are existing models, this weakens the novelty of this paper.
--- Lack comparison with sota image-text retrieval methods on image-text retrieval datasets in experiments.

**Questions:**

See weakness

**Limitations:**

See weakness

---

> ### Author Rebuttal · Authors · 2023-08-09
>
> Dear Reviewer **6Lfv**, thank you for your insightful feedback!
>
> **Lack comparison with SoTA image-text retrieval methods on image-text retrieval datasets in experiments:**
>
> Please see the global part of the rebuttal and the figures in the accompanying pdf, where we address this concern.

---

### Author Rebuttal · Authors · 2023-08-09


Dear Reviewers and ACs,
We were happy to see that the reviewers have found that our paper presents: “interesting problem and proposes an interesting setup” (L59s), “process with dialogue is interesting and largely under-explored” (34K9), The strength of this submission lies in its clear and compelling motivation (Z6DK), The proposed framework is practically valuable and well evaluated (6Lfv). Your input is instrumental for improving our paper.



The main concern of the reviewers seems to be regarding lack of comparisons with existing Text-to-Image retrieval baselines and evaluations on additional datasets. We would like to stress that we indeed compare our retrieval method to an existing Text-To-Image (TTI) retrieval baseline (on VisDial), reporting results in lines 174-176 and 208-210. More specifically, the baseline we compare to is BLIP, since this is the publicly available SoTA model for TTI [1]. For a fair comparison we further fine-tuned this baseline model on the VisDial dataset for the TTI task (providing it with images and their captions only). We find that the retrieval performance of the fine-tuned single-hop text-to-image baseline, is nearly identical to our dialogue-trained model (63.66% vs. 63.61%). This corresponds to dialogues with 0 rounds in Figure 2a. However, using increasingly longer dialogues in our method, eventually achieves retrieval performance over 81% (Fig. 2a), showing a huge improvement over the single-hop TTI SoTA baseline.



As for the choice of dataset, since VisDial is the only dataset containing image-dialogue pairs, we found it to be the most suitable dataset and benchmark for evaluating our method. However, due to the raised concerns and for further validation, we generated two synthetic image-dialogue datasets (using ChatGPT as a questioner, and BLIP2 [2] as an answerer) from two Text-To-Image (TTI) benchmarks, Flickr30k and COCO (see figures in the PDF attachment). We then compared our method to two TTI baselines, namely CLIP and BLIP, in a zero-shot setting (i.e., none of the compared methods have been fine-tuned on either dataset). We find that our method surpasses the two baselines by a large margin, on both datasets. Furthermore, when we provide the baselines with the concatenated text of the dialogues, instead of just a caption, they exhibit a significant improvement in success rate (please see attached figures) over the single-hop TTI attempt. Nevertheless, the gap in favor of our method is maintained (COCO) or increased (Flickr30k).

Our zero-shot results on COCO and Flickr30k (in attached figures) show that:

1.  Dialogues improve retrieval results for off-the-shelf TTI models. Although the CLIP and BLIP baselines have only been trained for retrieval with relatively short (single-hop) text queries, they are still capable of leveraging the added information in the concatenated Q&A text. Note that CLIP becomes saturated at a certain point due to the 77 token limit on the input.

2.  Our strategy of training an Image Retriever model with dialogues (as described in Sec. 3) further improves the retrieval over the compared methods, raising accuracy from 83% to 87% at single-hop retrieval, and surpassing 95% after 10 dialogue rounds (on COCO).


We believe the above addresses the main concerns in the reviews, and we welcome further discussion.

References:

[1] Junnan Li, Dongxu Li, Caiming Xiong, and Steven C. H. Hoi. BLIP: Bootstrapping Language-Image Pre-training for Unified Vision-Language Understanding and Generation. In ICML, pages 12888–12900, 2022

[2] Junnan Li, Dongxu Li, Silvio Savarese, and Steven C. H. Hoi. BLIP-2: Bootstrapping Language-Image Pre-training with Frozen Image Encoders and Large Language Models. CoRR, abs/2301.12597, 2023

---

### Decision · Program_Chairs · 2023-09-21

**Decision:**

Accept (poster)

**Comment:**

The work is about an approach to chat-based image retrieval, leveraging a language model for user interactions and aiming to enhance search results through dialog. The motivation for this research direction is clearly articulated, emphasizing the potential of chat-based interactions to refine search queries and improve image retrieval relevance.

However, reviewers pointed out some weaknesses in the paper.
- The lack of baseline comparisons to traditional single-hop text-to-image retrieval methods using the same dataset raises questions about the necessity of introducing conversation into the retrieval process. Such comparisons would better justify the complexity of the proposed approach and its potential advantages. Even though the authors mentioned they compared with BLIP in a few lines, the details of how BILP baseline was set up are missing, as well as some other baselines (e.g., BILP-2).
- The paper would benefit from more detailed explanations regarding the types of dialogues needed to aid in visualizing captions and the rationale for employing conversation in image retrieval.

Regarding datasets to be evaluated, it worth checking multi-modal dialogue datasets as well, for examples, I would recommend instead of simulated dataset in the future version, the authors can check some of these datasets for experiments (https://github.com/ImKeTT/Awesome-Multi-Modal-Dialog).